# Adolescents’ Perceptions of a Relapse Prevention Treatment for Problematic Gaming—A Qualitative Study

**DOI:** 10.3390/healthcare11172366

**Published:** 2023-08-22

**Authors:** Sevtap Gurdal, Sabina Kapetanovic, Isak Einarsson, Karin Boson, Emma Claesdotter-Knutsson

**Affiliations:** 1Department of Behavioral Studies, University West, 461 32 Trollhättan, Sweden; sevtap.gurdal@hv.se (S.G.); sabina.kapetanovic@hv.se (S.K.);; 2Department of Psychology, Stockholm University, 113 47 Stockholm, Sweden; 3Department of Clinical Sciences, Lund University, 221 84 Lund, Sweden; isak.einarsson@med.lu.se; 4Region Skåne, Child and Adolescent Psychiatry, Regional Outpatient Care, Lund University Hospital, 221 85 Lund, Sweden; 5Department of Psychology, Inland Norway University of Applied Sciences, 2318 Lillehammer, Norway

**Keywords:** treatment, problematic gaming, gaming disorder, interviews, children

## Abstract

Given the increasing prevalence of problematic gaming, in 2013, the diagnosis “Internet gaming disorder (IGD)” was included in the Diagnostic and Statistical Manual of Mental Disorders, 5th edition (DSM-5) as a potential diagnosis. With a new diagnosis, it is important to determine treatment options. The importance of the parent–child relationship has been emphasised in problematic gaming and its treatment. This study aims to provide more knowledge about adolescents’ perceptions of a treatment for problematic gaming and understand whether such treatment may have a bearing on the parent–child relationship. We conducted individual interviews with nine adolescents who completed a treatment for problematic gaming. The interviews were analysed using thematic analysis. The analysis revealed three themes. Theme 1: adolescents’ experiences of the new treatment; Theme 2: adolescents’ perceptions of the effect of the treatment on their gaming behaviour; and Theme 3: adolescents’ perceptions of changes in their parent–child relationships. The adolescents viewed the treatment as a way of gaining control of their gaming, a process in which a therapist played an integral part. For the majority of the adolescents in our study, the main effects of treatment were gaining insight into how their gaming and gaming-related behaviours affected other parts of their lives. The participants felt that the treatment improved their relationship with their parents through reducing everyday conflicts. This new knowledge can be used for the development of future interventions involving children and adolescents.

## 1. Introduction

Today, playing computer games is a common leisure activity for both children and adults in Sweden. According to a recent report from the Public Health Agency of Sweden, 49% of adult men and 37% of adult women in Sweden stated that they had played computer games in the last 12 months, with a total of 13% of adults having played computer games on a daily basis [1]. The percentages are even higher among children, with more than half of Swedish children between the ages of 9 and 18 stating that they play computer games daily [2].

Gaming is an important part of children’s and adolescents’ lives. In an Italian study, it was revealed how gaming can motivate individuals to strive for new goals and make progressions that are not limited to the game; the children that played a game were shown to report higher levels of engagement, internal locus of control, risk perception, and protective behavioural intentions.

Although for many children, gaming is a common leisure activity, some children may go overboard with the activity, which can cause problems in their regular daily lives. The prevalence of problematic gaming differs highly as different studies have used different measuring instruments and are conducted in different cultural contexts and ages, with prevalence figures ranging from 0.5% to 15% [3,4,5]. Globally, the prevalence has been estimated to be approximately 3%, with the highest numbers found in adolescents [6].

Risk factors for problematic computer gaming are male gender and youth [6]. There is a high comorbidity between problematic gaming and other psychiatric diagnoses such as neuropsychiatric developmental disorders, anxiety, depression, social phobia, and obsessive–compulsive disorder (OCD) [7,8,9,10]. Studies indicate that 32% of patients with problematic gaming also meet the criteria for depression [11]. Studies have also found a high degree of suicidal thoughts in the group [12]. It is common for problematic computer gaming to be associated with sleep disturbance [13] and poorer academic results [14]. In several studies, it has been shown that individuals with problematic computer gaming often feel lonely and/or have a lower social ability [14]. Moreover, problematic computer gaming correlates with difficulties in recognising and regulating one’s emotions [14,15].

In 2013, the diagnosis “internet gaming disorder (IGD)” was included as a potential diagnosis in the Diagnostic and Statistical Manual of Mental Disorders, 5th edition (DSM-5) [16]. The diagnosis consists of nine criteria, at least five of which have to be met with conditions lasting for ≥12 months in order for the person to be diagnosed as having IGD. The criteria are (1) preoccupation with computer gaming; (2) abstinence when gaming is removed; (3) tolerance to gaming, i.e., the need to play more exciting games; (4) fruitless attempts to stop playing games; (5) loss of interest in former leisure activities; (6) continued playing despite negative consequences; (7) denial to others of extent of gaming; (8) emotion regulation—escape from negative emotions; and (9) loss of a relationship or problems with school/work because of gaming.

The 11th revision of the International Classification of Diseases (ICD-11) included the diagnosis “gaming disorder (GD)” in 2018 [17]. According to the ICD-11, GD is defined by a persistent pattern of gaming behaviour characterised by (1) impaired control over gaming; (2) prioritising gaming over other activities; (3) continuation of gaming despite negative consequences; and (4) continuation of gaming despite gaming having a negative impact on the life situation, family relationships, employment, and other important areas of life. As stated in the DSM-5, the problems must have lasted for at least 12 months [17].

To address the troubles related to problem gaming, development of preventive or therapeutic measures is key. Although there is no golden standard treatment for IGD, most treatment studies for problematic gaming are variations of cognitive behavioural therapy (CBT). CBT is thought to help addicts identify self-defeating thoughts that may drive addiction. CBT aims to change negative patterns of thinking so that the outcome or choice is something other than video games [18]. Indeed, CBT is today a promising form of treatment for problematic computer gaming [6,18,19], with the majority of treatment studies being conducted in Asia [19]. Hence, more studies are needed in a Western context.

In a systematic review and meta-analysis of twelve treatment studies including a total of 508 patients aged 12–57 from seven countries, it was shown that CBT was strongly related to reduced symptoms of IGD (g = 0.92) and depression (g = 0.80); and moderately related to reduced anxiety (g = 0.55) after the treatments [6]. Well in line with this study, Kim et al. (2022) [19] concluded that psychological treatment including a combination of CBT and family interventions was effective in reducing problem gaming. As gaming is related to family relationships, the authors suggest that it is likely that a treatment with focus on the communication and relationship pattern between parents and children would be effective in treatment of gaming problems in children [19]. Relapse prevention (RP) is a CBT-based treatment developed to treat alcohol problems in adults, but the method is also used to treat addiction to alcohol, drugs, tobacco, and gambling among both adults and adolescents [20,21]. We developed a RP protocol/manual derived for treatment of child and adolescent IGD. Together with experienced clinical psychologists, the manual was adjusted to better suit children and adolescents.

Children and adolescents are part of social networks, such as parents, peers, and schools, within which they develop. As the family is the proximal socialising area in child development [22], adolescent gaming behaviour is inevitably embedded in the family and in the adolescent’s interaction with their parents.

Expecting parents to shape their child’s or adolescent’s gaming behaviours without acknowledging the child’s own role in their behavioural development and in the parent–child relationship may be a faulty expectation. Indeed, parents and children are engaged in a mutual chain of actions and reactions where children’s behaviours, feelings, and actions also have an impact on their parents [22]. As such, many parenting behaviours and practices relating to mediation of child gaming change as a result of their child’s gaming behaviours and frequency of gaming [23]. For example, problematic gaming behaviours seem to elicit more restrictive parenting practices from parents, which, in turn, may exacerbate the parent–child conflict as well as the child’s ability to control their gaming behaviours [24]. Similarly, close family bonds and cohesion between parents and their children are reciprocally related to adolescent gaming problems [25]. This suggests that changes in adolescent gaming behaviours may have an effect on the parent–child relationship and interactions. Such notion may be of particular relevance in terms of treatment of adolescent gaming problems and IGD. It is possible that individualised treatment would not only have an effect on the behaviour of the child, but the treatment could also have a spillover effect on the family’s interactions and parenting practices. Inevitably, family engagement in child therapy, as well as the support provided to the child during therapy, is critical to the effect of the treatment [26]. In this sense, the treatment and the changes in child gaming behaviours at home could in turn ease the interactions between parents and their children and induce changes in parenting practices and understanding relating to child gaming behaviours. To what extent children experience changes in parent–child relationships after undergoing treatment for gaming problems has yet to be understood.

Therefore, to gain more insight into adolescents’ perceptions of a treatment for problematic gaming and understand whether such a treatment may have a bearing on the parent–child relationship, we conducted individual interviews with adolescents who completed a treatment for problematic gaming. As suggested by the American Psychological Association (APA)’s Presidential Task Force on Evidence-Based Practice [27], qualitative approaches provide integral knowledge about practice involving people and, more specifically, children in clinical care. In this sense, the interviews conducted in the current study will provide more knowledge about children’s perceptions and voices, both in terms of their experiences of treatment and in terms of their parent–child relationships—knowledge that can be used in the development of future interventions for problematic gaming involving children and adolescents and their parents or caregivers. Indeed, children are individuals in their own right; at the same time, they are an inherent part of a family, and their caregivers are legal guardians who are often directly or indirectly involved in the treatment of children. In that sense, treatment of children could be seen from a holistic point of view, where both the child and, indirectly, the context of the child are informed. Giving children the possibility to voice their experiences and perceptions of the delivery of the treatment and its effects on their gaming problems has bearing on understanding the specifics of the treatment. In addition, knowledge of children’s perceptions of parent–child relationships—thus, the context they are embedded in—can be used for development of future interventions involving children and adolescents and their parents.

The following research questions guided our study: (1) What were the adolescents’ with reported problematic gaming perceptions of the RP treatment? (2) What were adolescents’ perceptions of the effect of the treatment on their gaming behaviour? and (3) What were adolescents’ perceptions of potential changes in the parent–child relationship as a result of the treatment?

## 2. Materials and Methods

### 2.1. Participants

Participants were recruited from a larger sample from child and adolescent psychiatric clinics (CAPs) in southern Sweden. The larger sample is from a randomised control study evaluating an individual treatment for problematic gaming in adolescents. A total of 622 adolescents aged 13–18 who had their first visit to the CAP were screened for their gaming problems. Adolescents who met the proposed DSM-5 criteria for IGD [16] were offered participation in the treatment study consisting of seven to nine sessions of 45 min over a period of 7–9 weeks. Among the CAP patients meeting the criteria for IGD during the study’s inclusion period, a total of 113 patients agreed to participate. One patient was excluded because of incorrect inclusion, being younger than 13, and ten patients were excluded as a result of not completing the follow-up measures. The final sample consisted of 102 participants aged between 13 and 18 years.

In the next step, adolescents were randomised into two groups: a treatment and a control group. For randomisation, we applied a random allocation sequence using the “chit method” by preparing chits of paper indicating either control or treatment [20,21]. Participants were randomised in a 1:1 ratio to either intervention or control. A total of 47 adolescents underwent the treatment and 55 were allocated to the control group.

Purposive sampling was used, and twelve adolescents from the treatment group were randomly asked to participate in the interview study with a wish to include at least two girls. The only inclusion criterion used was that the participants had undergone the treatment. The exclusion criterion was being allocated to the control group.

Ten agreed to participate, but one dropped out as he did not respond to emails.

A total of nine adolescents agreed to participate in the interviews: seven boys and two girls 13–17 years old. Table 1 presents a summary of their specific ages, gender, gaming habits, and treatment history.

All participants received written and verbal information about the study and provided written consent, according to the Swedish Act concerning the Ethical Review of Research Involving Humans (SFS 2003:460). Adolescents under the age of 15 years also provided their caregivers’ consent.

### 2.2. Intervention

Based on the theory of relapse prevention (RP) for alcohol and substance abuse [28], we developed a manual for problematic gaming together with four experienced psychologists in the field. The treatment was offered to children and adolescents both in person and via a video link to facilitate participation and consisted of three parts: (1) setting goals; (2) understanding and identifying high-risk situations and problem behaviours; and (3) consolidating a new activity schedule and identifying future high-risk behaviours. Clinicians (psychiatrists and psychologists) were trained in CBT beforehand and supervised by a clinician with vast experience in treating adults with gambling and gaming addictions. For more information, see André et al. [20] and Kapetanovic et al. [21].

### 2.3. Procedure

The empirical data were collected through individual semi-structured interviews [29]. Open-ended questions were asked to obtain the participants´ experiences and perceptions of the treatment as well as their perceptions of changes in their relationship with their parents as a result of the treatment. Interview guides were discussed and developed by all the authors together to make sure that we captured the aim of the study. The interviews were conducted 6 months after the end of the treatment. The first and the third author (S.G. and I.E.) and two assistants conducted the interviews digitally, by phone, or face to face. Each interview lasted 15–35 min and was recorded either with a recorder or by Zoom. The interview guide was focused on the following themes: adolescents’ experiences of the treatment and the effect of the treatment concerning their gaming behaviour as well as their perceptions of their parent–child relationship (please see Appendix A).

### 2.4. Data Analysis

All interviews were transcribed verbatim and then analysed by the first author (S.G.) and the last author (E.C.-K.) using thematic analysis. The study explores a new area, trying to find out how adolescents with reported problematic gaming perceive a treatment for problematic gaming; the study used an inductive approach that was more explorative and not generalisable with the small sample used. The interviews and results will help to change and develop new strategies for treatment of problematic gaming or gaming disorders.

The analysis followed the six steps recommended by Braun and Clarke (2006) [30] and had the research questions in mind. In the first step, the first and the last author separately read all the interviews to familiarise themselves with the material and get a sense of the content; they made notes and highlighted parts of interest to the study. This step is a close read of all material where every word is important. The second step was to generate initial codes at a semantic level, such as “difficulties with the treatment” or “forgot the assignments”, staying close to the participants’ own descriptions. From this step onward, the analyses were conducted jointly by the two authors involved. In the next step, we started searching for themes that were related to the research questions. The codes were read through to find similarities between them for grouping into themes or subthemes. Step four was about reading through all themes and subthemes to check the consistency of the themes and the themes’ relation to each other. In step five, we checked for any overlaps and merging of themes or subthemes. The final step was to present the themes and subthemes in a relevant and understandable way. In our case, we used the research questions as a basis for the presentation. After the final step, the other authors checked the themes and subthemes, and the research group discussed them together to confirm reliability. Some subthemes were collapsed or moved to another theme. Further, when analyses were finished and discussed in the research group, the decision that no more interviews were needed was decided upon.

Extracts from the interviews were used to give the reader a clearer picture of the themes and were translated by the second author (S.K.). In order to maintain conformability, the researchers have strived to be close to the transcribed text using quotes when presenting the results.

## 3. Results

The analyses of the interviews revealed several themes and were related to the interview aims and interview guide. That is, theme 1 relates to the adolescents’ experiences of the new treatment; Theme 2 relates to the adolescents’ perceptions of the effect of the treatment on their gaming behaviour; and Theme 3 refers to the adolescents’ perceptions of changes in their parent–child relationships as a result of the treatment. Each theme also includes several subthemes (see Table 2).

### 3.1. The Adolescents’ Experiences of the Treatment

Relating to the first theme and how the adolescents perceived the new treatment for problematic gaming behaviour, one issue the adolescents mentioned was the assignments they were told to complete between the sessions. Some of these were positively perceived and some were more negatively perceived. This theme is divided into three subthemes: strategies for less gaming; thoughts about the home assignments; and relations with and importance of clinicians.

#### 3.1.1. Strategies for Less Gaming

In the interviews, the adolescents described the treatment and discussed different strategies for less gaming. For example, having goals was one of these strategies. One girl described that it was a good approach to set goals and then follow-up on the results. She had to think about how the gaming affected her, her schoolwork, and her eating. Having goals made her more aware of the problem and how much gaming impacted her life.


*After all, it wasn’t bad to set goals, to sort of discuss, and both set goals and discuss how my gaming affected may daily life. I don’t remember now, but we wrote down a lot of different things that were affected, like that I … didn’t do my, I didn’t take care of schoolwork, I didn’t do my homework, I didn’t eat very well, in a sort of way I already knew before: it can’t go on like this. But I didn’t realise how much was affected.—Girl 7*


Another participant explained that his goal was to learn how to be calmer during the game as well as to reduce the time he spent on gaming.


*Yes, but I suppose the goal [of the treatment] was that I should be calmer when playing and, like, not play as much. I suppose I thought about that sometimes when I was playing, and then I calmed down. Just having that thought in your head, that you go there [to treatment] to reduce your gaming makes you think while playing that “OK, I suppose I should be calm since I am in treatment.” Yes, I think it has helped me to see how I play, how I should play and what makes me angry—and how to think in order not to get angry.—Boy 6*


Another strategy for less gaming was to control the amount of time spent gaming. Since gaming was a frequent activity which took a lot of the adolescents’ time, there were different ways of taking control of gaming and decreasing the time spent on gaming that were suggested in the treatment. The adolescents described it in different ways. For example, one boy expressed the importance of not using gaming as a distraction and said that this was one of his main goals in the treatment. Instead of gaming, he had to think of something else to use as a coping strategy or, as he described it, a distraction.


*You shouldn’t use it for distraction; there are other things you can do. I suppose that was the biggest goal we set.—Boy 1*


Control was also mentioned as something that is not only related to a total lack of gaming, but also appeared as thinking of gaming in a balanced way. For instance, the adolescents could avoid going online every time that a friend suggested gaming.


*Control my gaming? Mm, well, not control but perhaps, like, balance my gaming. Before, I just gamed, let’s say if my mate said, “Shall we game?” then I just said “OK” but now it is “I must do my homework first.”—Boy 2*


Taking control also seems to include the possibility to choose when to game or when not to game, as one boy put it. Another strategy was to actively plan pauses from gaming. The adolescents were encouraged to plan their gaming time. Part of this involved planning pauses. Since several in our study of the adolescents had difficulties keeping track of how much time they spent on gaming, this was perceived as a helpful strategy.


*Having these pause periods without gaming, that did help, because it made me realise that … that the gaming actually didn’t do anything for me; I am just as happy without it.—Boy 3*


The same participant continued to describe how the planning helped him realise that gaming is perhaps not as important as he had imagined.


*That I was to try to spend more time with my mates outdoors and away from the gaming. That I would try out, like, going without it on certain holidays or certain days and see how it goes, and if it works well, and if it worked well for me not to play games for, let’s say, a week, or a period of time, then we extended it to 2 or 3 weeks … Personally, I believe that I have changed my gaming myself, with the help of my own way of thinking. The treatment planted the seed into my head.—Boy 3*


According to the next participant, reflecting and not just starting a new game was a good way of continuing to game after one match was over. The treatment made him learn to stop and not begin a new game right away.


*Like, getting ideas about how to turn it off, so that I don’t begin a new match when I have 5 min left of my game time.—Boy 4*


#### 3.1.2. Perceptions about the Home Assignments

The treatment included home assignments for the adolescents. These assignments included writing a gaming scheme, trying to replace gaming with something else, taking control, etc. The participants had different things to say about the assignments. Some saw the assignments as helpful; others were more negative. The school-aged children in the treatment group said that the assignments reminded them of schoolwork and that they were boring. One boy said it was like having homework and that he already had a lot of that and therefore was not that eager to complete it.


*It was more that it felt like homework, I already had homework to do and then I had this assignment as well, so it felt kind of like homework. Then, well, it takes away a bit of time out of my day so I thought, I don’t want to do it [laughs].—Boy 2*


He further said that the assignment to make a gaming scheme was OK, although he did not have the energy for it.


*One thing I thought was a bit troublesome, or how to put it, I was given pieces of paper and so on, I was to write, like on the first or second day, I was to make a gaming chart, like draw it, when I play; it was OK but it was kind of like getting homework, and you know I can’t be bothered, if I have like one piece it’s OK but if I have four or five I can’t be bothered doing them.—Boy 2*


An opposite thought was revealed when one adolescent was told to replace gaming with something else. His assignment was to bake a cake, which he described as a positive experience.


*Yes sure, they were interesting, like “Make a cake.” There were many different things to think about, there were different tasks to think about, what you want to change and, yes, there was a lot of interesting homework.—Boy 6*


It also happened that the adolescents had forgotten their assignments for the next session and then there could be no follow-up.

#### 3.1.3. Relations with and Importance of Clinician

A clinician followed the adolescents throughout their treatment and became an important person who developed a relationship with them. Although some of the sessions were held in the office, others were held online. The adolescents expressed that some of the clinicians had knowledge about gaming, which was highly appreciated by the adolescents. One boy said,


*Yes, I thought it was much more fun, because he understood what I was talking about and what I meant, so, yes, that made a big difference.—Boy 5*


Another boy said that it was easier if the clinician was a gamer because then the clinician could relate to him when he explained what it was like to game. This could be an advantage in the treatment.


*I would have thought that, surely, it’s easier if you’re a gamer yourself as a clinician, because then you can recognise more things than if you don’t play games at all, but then it’s also good to find out how people behave so that you get a more general idea of what most other people do rather than just you yourself and one other person.—Boy 6*


Some knowledge about the games appears to have sped up the beginning of the treatment as the adolescents felt they could start talking without having to give explanations. The participants mentioned how the clinician’s knowledge made it both easier and safer to talk about their gaming.


*He played a game that’s the same as the one I play, so he, like, knew. If I was to mention the game then he knew, like, what I was talking about. It felt easier to talk with him. If I was to talk about the game with you, it would be, like, difficult, to explain things, like, because you don’t know what I’m talking about. He knew what I was talking about.—Boy 2*


Besides the positive thought about the clinician’s gaming knowledge, the adolescents mentioned that it was good just to talk to someone about their gaming habits. This was a new person to talk with, and here was a new opportunity to talk to someone who understood them.


*It was quite good to be able to talk about what I was feeling; I thought it was a relief to get things out that I’ve been wanting to say for several years, and then it was of course very, I really appreciated this help. To get things out that I wanted to say, I would say that was a very good feeling, because I have held those thoughts back for a very long time.—Boy 5*


### 3.2. Effect of the Treatment

The follow-up interviews showed some effects of the treatment, especially concerning a changed view about gaming habits and how some of the adolescents had started to do other things in place of gaming. The subthemes are changed mindset about gaming; and time for other activities.

#### 3.2.1. Changed Mindset about Gaming

Some of the adolescents started to think about their gaming habits during and after the treatment. They expressed that they knew, to some extent, that too much gaming was probably not good for them, but they had not put this thought into words before. One boy described how the treatment had made him see clearer and how he had started thinking about the consequences gaming might have.


*Once again, it was always in my mind that it was something negative. This treatment sort of made me see this more clearly and think about the reasons—I mean, the negative effects gaming can have on my life.—Boy 3*


This was expressed by another boy:


*It’s that you talk about the drawbacks, you talk about it. It’s something I hadn’t thought about before the treatment.—Boy 1*


The new insights and consequences of gaming were pointed out by others, and some adolescents expressed that they had not thought about gaming as problematic or as having bad consequences for future life. One boy described that he had been so “into” gaming without reflecting on the drawbacks it might have.


*Well, I don’t know what I thought before, I sort of didn’t think about—sort of that it was problematic or something; now I understand that it’s bad to be so deep into it and that you should sort of not take it so seriously, and so on.—Boy 6*


There were also some participants who largely stopped gaming after the treatment, as expressed in the following:


*I finished the treatment perhaps a month ago and I have not been gaming a great deal since then. I’ve been away and I’ve had a summer job.—Boy 7*


#### 3.2.2. Time for Other Activities

Some of the adolescents commented how the treatment had opened their eyes to other activities instead of gaming, mainly schoolwork. As a result of talking to the clinician, one boy realised that, instead of always prioritising gaming, he should be doing his homework.


*If you think about it by yourself in your own head it doesn’t work very well, but if you say it out loud to someone else then you understand it better, like … and then I sort of reflected on it, like, OK perhaps I shouldn’t prioritise the gaming above this—above homework. Maybe I should do my homework as soon as I get it instead of procrastinating.—Boy 2*


One of the girls said that she wanted to invest in school and try to improve her grades:


*I want to prioritise school a bit and try to make more of an effort to sort of, actually be a bit better than just passing.—Girl 7*


The same girl also said that she wanted to take hold of her life, which the treatment made her realise.

### 3.3. Changes in the Parent–Child Relationship

The adolescents described different changes in their parent–child relationship as a result of the treatment. They experienced fewer quarrels and conflicts with their parents and described spending more time with their parents now than before the treatment. The following subthemes are presented below: less conflict; spending more time with parents; and parents’ changed attitude to gaming.

#### 3.3.1. Less Conflict

Fewer conflicts and quarrels were described by several adolescents in the interviews. Some experienced smaller changes, while for others, change was much more evident. Having had major conflicts in the past, they described that these had reduced considerably and that less gaming had led, and continued leading, to fewer arguments.


*Well, it was really, really a lot before the treatment but now, ever since the treatment, it has happened perhaps once, twice; before the treatment it was every day. Yes, and it feels very good, we have a better home climate, the relationship in the house, it’s got much better. It was more that I, they turned off the internet and then I got annoyed and then it was that I had stayed up too late, they could see that I had stayed up all night, and so on. And when they took away my devices there was an awful lot.—Boy 4*


Boy 4 also described that previously, when he had spent a lot of time on gaming, there had been quarrels about him being up too late; his parents would confiscate his consoles, which, in turn, had exacerbated the conflicts. Boy 5 stated that his goal now was to game less, that this would affect his relationship with his parents, and that it had already helped.


*I was to do less gaming and I would have a better relationship with my parents, listen to them more when they tell me to stop gaming, that is what I remember. It has actually got better.—Boy 5*


As well as having fewer arguments about gaming, there are examples of how the cause of arguments had shifted. One boy related those arguments about gaming had been replaced with other arguments.


*Yes, I’ve had quite a few arguments with my Mum during the year I’ve lived with her, and most of them have been specifically about my gaming, that I don’t do anything else and that I waste my time doing it. There is less arguing about the gaming now, since I don’t game but instead, we argue about other things, so these arguments swap places with the gaming.—Boy 3*


#### 3.3.2. Spending More Time with Parents

The insight that the young people gained through treatment seems to have influenced their willingness to spend more time with their parents. One boy expressed that he thought that the change was positive and that he now spent more time with his parents.


*I think it has changed in a positive way, I spend time and chat with them more than I did earlier.—Boy 3*


Another boy expressed that it was easier to spend time with his family now compared with before, when he would just leave his room occasionally to join them downstairs.


*Once I get downstairs, we look at a film or a series—I chat with my brothers, the same with my parents, I can have a chat with them without feeling that I need to go away.—Boy 5*


One of the girls related that after the treatment, she would rather spend an extra 20 min with her mother than game for 20 min.

#### 3.3.3. Parents’ Changed Attitude to Gaming

According to the adolescents, some parents had, during the treatment, changed their attitude to gaming. For example, one talked about how his mother had changed:


*She probably understands what I do better [referring to the change].—Boy 2*


Boy 5 described his mother as less restrictive than his father, but the father too had become more positive after the treatment.


*I know that my Mum has sort of looked at it in the same way, she hasn’t been massively negative, unlike my dad who was super anti earlier. But I suppose my dad has become, he has almost tried to encourage me to play games, because I haven’t been very cheerful recently, he has, like, gone, “Won’t you plug in the computer now?”.—Boy 7*


The father seemed to have related to the boy’s change in stance toward gaming and tried to persuade him to game. It appears that the treatment may have changed his attitude to gaming and his perception of why his son gamed.

## 4. Discussion

Children and adolescents spend a lot of their time gaming. For many, gaming is a daily activity [1,2]. For some children and adolescents, gaming can be addictive, which is why IGD was included in the DSM-5 as a potential new psychiatric disorder in need of more research [16]. There is as yet a lack of clinical research as well as treatment for problematic gaming, especially among children and adolescents. To address this knowledge gap, we developed a RP treatment protocol/manual for child and adolescent with IGD [20,21]. Wanting to emphasise child and adolescent voices and lived experiences [27], we conducted individual interviews with adolescents who completed the treatment. Our aim was to understand their views about gaming, as well as about the RP treatment for problematic gaming and whether they perceived that the treatment had any effect on their relationship with their parents. Such knowledge is essential for development of future treatments involving children and adolescents.

### 4.1. Adolescents’ Experiences of the RP Treatment

Overall, the experiences of the RP treatment were focused on regaining control of gaming-related behaviours—such as planning for breaks, planning for other activities, setting goals, and creating a gaming schedule. This can be understood in light of previous research that has found that adolescents with problematic gaming have difficulties regulating their emotions [15]. Gaming can be used both as way of dealing with adverse events—a coping strategy—and as a fun leisure activity. It may be speculated that because gaming is used in both ways, for coping and leisure, several of the adolescents in our study are unaware of the way they use gaming to regulate their emotions and instead only view it as a fun way to spend their spare time. The RP treatment seems to have helped the adolescents to understand gaming and how to control both gaming and their emotional expressions while gaming.

The adolescents seemed to appreciate talking to someone about their gaming in a way that was usually not possible with their own parents. This finding is in line with previous qualitative studies about the importance of creating a motivating and supportive environment in IGD treatment for youth [31]. Talking about their gaming-related behaviour and thoughts was found to be helpful by the adolescents, particularly if the clinician had an understanding of gaming or had experiences with gaming as an activity. It was found to increase their motivation for treatment and, as suggested below, their understanding of the negative consequences of their gaming-related behaviour.

The RP treatment appears to help affected adolescents regain a sense of control of their gaming-related behaviour, a core criterion in both the APA [16] and WHO [17] conceptualizations of problematic gaming. Several of the adolescents described the homework—an important aspect of RP treatment [32,33] and CBT treatments in general—as boring, reminding them of schoolwork, and said that they had difficulties doing it. Future work should strive to adapt and adjust the content of the homework to be engaging and meaningful for the adolescent being assigned it.

### 4.2. Effect of the RP Treatment

For several of the adolescents in our study, one of the main benefits of the RP treatment was that it made them aware of their gaming-related habits and the way these were affecting other parts of their lives—something that many of them had not thought of before. This may partially be a consequence of the sample characteristic where several of the adolescents in our study were referred for treatment through their parents and only a minority were self-referred. Therefore, the adolescents in the RP treatment were most often not aware of their problems and in turn were not highly motivated at the start of treatment. The adolescents who managed to complete the treatment, however, had a better understanding of the consequences of their gaming. Future RP treatments should consider both the shift in motivation of the individual and that adolescents with problematic gaming vary extensively in their view of the consequences of their gaming-related behaviours. Both factors have consequence for the affected adolescents’ motivation for engaging in a treatment aimed at helping them gain control of their gaming-related behaviours. Motivation is a core aspect of RP treatment [31,33], and this indicates that future treatments would do well to incorporate components that explicitly deal with patients’ motivation for treatment.

Moreover, based on the adolescents’ perceptions, the cessation of or abstinence from gaming may not be an appropriate goal for the RP treatment: only a few participants were willing to quit gaming entirely. The RP treatment could instead aim for a more balanced form of gaming, in line with Sim et al. [34]. A balanced way of gaming makes time for homework, socialising offline, and chores; but, more importantly, it also gives the adolescent the opportunity to develop other interests and activities. This might also facilitate engagement in treatment, especially with adolescents who might not be motivated to entirely quit gaming.

### 4.3. Changes in the Parent–Child Relationship

Most of the adolescents were referred to the treatment through their parents. As child behaviours and activities are embedded within a family, it is clear that parent–child interactions work in a mutual manner [22]. As parents have an impact on what their children do, children in turn have an impact on what their parents do and on their interactions with their parents. As parent–child conflicts tend to increase as a result of the child’s excessive gaming [24], we expected that adolescents who had completed the RP treatment would also experience a change in their parent–child interactions. Indeed, less conflict with parents emerged as one of the positive effects of the RP treatment. As shown in our results, for several of the adolescents in our study, gaming and gaming-related behaviours were a major source of conflict within the family, giving credence to the finding that close family bonds and cohesion between parents and their children are reciprocally related to adolescent gaming behaviours [22,25]. Future treatments might make the parent–child relationship a direct aim of intervention by involving parents or caregivers as a part of the treatment. We do, however, think that one-to-one sessions with adolescents are an important and integral part of the treatment, especially in relation to motivation, as is detailed above in the discussion about the adolescents’ perceptions of the RP treatment.

## 5. Limitations of the Study

This study has limitations that should be considered. Firstly, the results should not be interpreted as generalisable since not all of the adolescents who received the RP treatment were asked for a follow-up. Apart from age and male gender, commonly listed and established comorbidities are ADHD, anxiety, and depression. Our study was performed in a clinical setting where each of the participants was diagnosed with a psychiatric condition that might affect the generalizability of the results [20,21]. However, this specific circumstance could also be considered as strengthening the external validity since psychiatric comorbidity, not least ADHD, is a known feature of IGD. The majority of the adolescents in the study had had previous contact with child and youth psychiatry for reasons other than problematic gaming. Some of them had various forms of mental illness or functional impairment, with changes in symptoms over time, and were suggested RP treatment for problematic gaming during an ordinary appointment. This means that they did not apply for the RP treatment firsthand and that the motivation for the treatment may have been lower than if they had applied by themselves. Also, the current study did not include adolescents who did not complete the RP treatment, which could have impact on the credibility of the results. Future studies should strive to also include the perceptions of adolescents who only partly completed the treatment to get a broader picture of the intervention of interest. Moreover, we analysed the data by ourselves, which might have affected the objectivity. Finally, the current study has a qualitative approach with a small sample, which limits the generalizability.

Despite these limitations, the results of the current study are novel and timely and can be used to inform further development and changes concerning treatment for problematic gaming. As treatments for problematic gaming or gaming disorder are not well developed, particularly for children and adolescents, our study contributes with important information about how this group experiences a RP treatment for this condition.

## 6. Conclusions

The present study was undertaken to examine how adolescents who had undergone a RP treatment for problematic gaming experienced the RP treatment and whether they felt that the treatment had had an effect on their parent–child relationships. We found that the adolescents viewed the RP treatment as a way to gain control of their gaming, a process in which the clinician played an integral part. Moreover, for several of the adolescents in our study, the main effects of RP treatment were to gain insight into how their gaming and gaming-related behaviours influenced other parts of their lives; they felt that it improved their relationship with their parents through reducing everyday conflicts.

The concern about children and adolescents spending more time on gaming and about some of them developing a problematic relationship with gaming must be taken seriously. Our interviews with children and adolescents in RP treatment for problematic gaming show the importance of working with motivational aspects of RP treatment, and this might need to be a particular focus in working with children and adolescents, as several of the adolescents in our study were referred for RP treatment by their parents. In line with this, we believe that clearly involving the parents in RP treatment and in the RP treatment plan may be an important next step in the RP treatment for this group; just how central the parent–child relationship is both in the children’s understanding of the negative consequences of their gaming-related behaviours and in the improvement a treatment can bring in this relationship have emerged. It may also be fruitful for clinicians to always view gaming in the context of the particular child or adolescent in front of them and how it affects their schoolwork, other activities and important relationships. This can contribute to understanding of the patient´s problem, which is vital for developing a RP treatment plan and setting goals; it may also be a way of creating motivation and engagement in RP treatment, in the child or adolescent. Transferability is possible and findings may be transferred to another setting, context, or group but with an important note—this study´s participants are recruited from a clinical sample. For future intervention, the importance of including parents or caregivers is important; in place treatment, at least for the first time, is preferable.

## Figures and Tables

**Table 1 healthcare-11-02366-t001:** Description of participants in the interviews.

Participants	Age, Yrs	Gender	Treatment Sessions	Gaming Habits
1	14	Boy	entire treatment in person	gaming on console and computer
2	17	Boy	entire treatment in person	gaming on computer
3	14	Boy	missed 1–2 sessions in person	gaming on console (but sold it after the treatment)
4	15	Boy	entire treatment online	gaming on console and computer
5	15	Boy	entire treatment online and in person	gaming on console and computer
6	13	Boy	entire treatment in person	gaming on iPad or Nintendo Switch
7	17	Girl	entire treatment in person	gaming on computer
8	13	Girl	entire treatment in person	gaming on smartphone or compute
9	14	Boy	entire treatment in person	gaming on smartphone and laptop

**Table 2 healthcare-11-02366-t002:** Themes and subthemes.

Themes	Subthemes
The adolescents’ experiences of the treatment	Strategies for less gaming Perceptions about the home assignments Relations with and importance of the clinicians
Effect of the treatment	Changed mindset about gaming Time for other activities
Changes in the parent–child relationship	Less conflict Spending more time with parents Parents’ changed attitude to gaming

## Data Availability

The data used in this study is available from the submitting author upon request.

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
