# Peer review of "Adolescents’ Perceptions of a Relapse Prevention Treatment for Problematic Gaming—A Qualitative Study"

_healthcare, 2023, doi:10.3390/healthcare11172366_

Round 1

Reviewer 1 Report

Gaming holds significant importance in the recreational and social lives of adolescents. Nevertheless, excessive gaming has been linked to negative outcomes in this age group. Studies have revealed that familial factors, including the quality of parent-child relationships, parental control over media usage, and the marital and socioeconomic status of parents, can influence the likelihood of an adolescent developing gaming-related problems. Several interventions for problematic gaming have employed family approaches or involved modification of family-related variables. In this context, the current paper appears to be interesting by exploring the adolescent’s perceptions on treatment for problematic gaming and its family influences. By doing so, it aims to provide valuable insights to inform interventions focused on dealing with problematic gaming issues.

 However, the manuscript would benefit mainly by bringing in more clarity in its goals and conceptual framework. Please see my comments below to strengthen the findings presented in this paper.

Abstract: Abstract should highlight the key themes that were evolved.

Introduction: In lines 119-120, the sentences appear incomplete. Future interventions for ???. Is it involving only children or their family also.

The current introduction lacks clarity in explaining the relevance of treatment for gaming problems in relation to changes in parent-child relationships. It does not clearly articulate the relevance of how family influences impact the treatment process or why understanding the changes in parent-child relationships after therapy is essential for informing interventions. To strengthen the introduction, more focus is needed to build a strong rationale for why understanding family influences and the effects of therapy on these influences is critical in informing effective interventions.

The research questions should be more specific and self-explanatory. For example ?? adolescents, is these adolescents with gaming disorders?  The introduction does not provide adequate rationale for research questions 2 and 3 and there is lack of coherence on how the rationale were presented for all the 3 interlinked research questions.

The introduction should also discuss the intervention available for gaming disorders and discuss how their interventions align with the existing ones. This is very important and this also should be reflected in the title of the study.

Methods: in lines 139-140, the authors should provide reference for randomized control study (at least trial registry number if protocol not published). The authors should clarify that whether their intervention and the intervention used in randomized control study are different or same?

The current format of presenting intervention before study participants is quite confusing and allow the readers to make their own assumptions. Thea authors can rearrange the flow for better clarity and avoid ambiguity.

The authors need to provide clear description of the inclusion and exclusion criteria to ensure more valid interpretation of the study findings and avoid ambiguous assumptions by the readers.  It was not clear why only 12 participants were allowed to participate? Who were those 12 and what were their eligibility criteria?

The study did not include adolescents who did not complete treatment. Missing out such negative or deviant cases would have impact on the credibility and maximum diversity sampling. The authors need to discuss this in the limitation section.

Similarly, mental disorders/problems usually co-occur/ co-morbid with gaming disorders. This should have impact on the context that are being studied. The authors need to bring this in discussion if co-morbid conditions were not addressed adequately during the study conduct.

The authors can provide information on the outcomes/improvements following the interventions to derive meaningful and impactful interpretations of the study findings.

Data analysis: the authors need to clarify whether the analysis is deductive or inductive.

The authors need to clarify what steps were taken to ensure trustworthiness (credibility, confirmability, transferability, dependability)

Other methodological aspects like data saturation, transcript returns etc should be mentioned. For the methodology, the authors could use the COREQ guidelines to improve their reporting and avoid redundancy

Results: Perceptions would be a better term for thoughts for 3.2.2 subtheme 2 for theme 1.

Throughout the manuscript, the authors should be specific about the interventions that was provided. The usage of term “treatment” throughout the manuscript particularly in discussion section should be avoided and replaced with the specific treatment that was provided. This is because the treatment is more generic and non-specific and can be referring to anything.

Similarly, the discussion should be made in relevance to the specific intervention that was provided.

A para on implications of the study findings in forming future interventions should be added.

Then interview guide should be provided in  supplementary appendix.

English language and grammar require revision at places throughout the manuscript.

Reviewer 2 Report

This paper focuses on really actual and interesting domains. In my opinion, its main strength is the connection between research and treatment, giving the opportunity to the second one to be improved thanks to the suggestions offered by interviews. I also think that the qualitative approach, even if only through a thematic analysis, is welcome in line with the overall aims.

However, several limitations are appropriately declared by the authors. As the authors emphasize the importance of parents in the whole process, a suggestion is to involve parents not only in the treatment but also in the related research.

However, I think the manuscript could be improved in accordance with the following suggestions:

a) Even if the proposed argumentations in the theoretical background are widely supported by scientific and psychological literature, it is widely acknowledged that gaming is a very complex reality and a various social phenomenon. Therefore, I think that a one-face argumentation concerning it could be restrictive. I suggest to insert some claims about the opportunities offered by online gaming. As an example, you can look at the article:

D’Errico, F., Cicirelli, P. G., Papapicco, C., & Scardigno, R. (2022). Scare-away risks: the effects of a serious game on adolescents’ awareness of health and security risks in an Italian sample. Multimodal Technologies and Interaction6(10), 93.

b) it could be enlightening to say something about why CBT is widely used in addressing the problem gaming

c) line 128: it could be useful to have some notions about the pinnacles of this theory

d) even if approved by ethical commissions, I think you should say something about the fact that just half participants were destinated to intervention: could also the “control” guys benefit from the treatments? Did you think to involve them in the treatment in a second time?

e) even if a little sample, did you find any differences concerning the different gaming habits?

f) line 196: I think the first sentence should be modified since the themes were related to the interview aims and guide.

g) so, in this stage the parents had no role during the treatment?

Minor revisions:

Line 70: the word ‘problem’ is repeated

Line 155: they or he/she?

Line 249: delete "of the adolescents"

The quality of English is good, in minor revisions you find some suggestions. 

Reviewer 3 Report

This is a well written paper. Details of the research project are reported. I really appreciated that the authors described the qualitative data and content analysis process in detail.

A few minor concerns are:

1. The qualitative data were analyzed by authors themselves, calling into question the issue of objectivity.

2. Some themes were identified from the data analysis. It would help to report the % of responses that contributed to each theme. I was also wondering if there were any responses not consistent with the themes. For example, did anyone report that there's "more conflict"? Or "spend less time"? It would be helpful to report those if possible.

Round 2

Reviewer 1 Report

The authors have addressed most of the concerns diligently.

For the benefit of readers, the authors need to include exclusion and inclusion criteria explicitly in their manuscript.

Similarly, the author's clarification of the concern regarding the steps that were taken to ensure trustworthiness (credibility, confirmability, transferability, dependability) should be included in the manuscript in the condensed form (avoiding duplication)  for the benefit of the readers.

English language and grammar can be rechecked and revised appropriately

Author Response

Dear reviewer 1,

Thank You for Your valuable comments. We have changed the manuscripte accordingly and hope You find that the mansucripte has improved. We have gone through the lagnuage spelling and grammar.

We have added inclusion/exclusion criteria see line 184-186. 

We have also rewritten parts of the manscripte inorder to make sure that the steps we have taken to  ensure trustworthiness is clearer.

Br 

Dr Claesdotter-Knutsson

Reviewer 2 Report

Thank you for the proposed improvements as well as for the appropriate answers. However, the article I suggested wanted to emphasize more the general opportunities concerning the adolescent 's world (from the cognitive, social and motivational side) rather than the specific research outcomes.  After this last suggestion, I think that the article has been generally strenghtened, thus it can be accepted 

Author Response

Thank you for your comment.

We have changed the sentence with the reference and hope it is more correct.

See line 40-43

Br Dr Claesdotter-Knutsson